# Co-developing a survey on public understanding of sustainable clinical research: A study protocol

Dylan Keegan *, Linda O'Neill ◉°, Peter Doran°

Clinical Research Centre, School of Medicine, University College Dublin, Dublin, Ireland

☻ These authors contributed equally to this work.
* dylan.keegan1@ucdconnect.ie

## Abstract

### Background

Clinical research is essential for medical advancement but contributes to carbon emissions through activities such as travel, data management, and laboratory processes. As healthcare systems move towards net-zero targets, there is a growing need to understand public awareness of and attitudes towards environmental sustainability in clinical research. Public and patient involvement (PPI) can enhance the relevance and acceptability of research, however there is limited evidence on how best to co-develop tools to assess public understanding of sustainable clinical research.

### Methods

This study is a two-phase protocol. Phase I involves the co-development of an anonymous online survey with PPI partners through two structured workshops. Survey themes, topics, questions and format will be agreed collaboratively, and the co-development process will be evaluated using qualitative data collected during the workshops. These data will be analysed thematically, using Braun & Clarke's (2006) framework. Phase II involves distribution of the survey to clinical research, patient and public networks using an online platform. Survey data will be analysed using descriptive statistics for quantitative data and inductive content analysis for qualitative responses.

### Discussion

This study will generate a co-developed survey instrument and provide insights into public understanding of sustainable clinical research. Reporting on both the survey findings and the PPI co-development process will inform future research and support the development of sustainable research clinical practices.

**Data availability statement:** No data are associated with this protocol.

**Funding:** This research is funded by the University College Dublin Clinical Research Centre (UCD CRC). The corresponding author, Dylan Keegan, received funding as part of a PhD programme. The funder had no role in study design, data collection and analysis, decision to publish, or preparation of manuscript.

**Competing interests:** The authors have declared that no competing interests exist.

## Introduction

Clinical research is essential for facilitating medical progress, through the development of new medicines, the generation of medical knowledge and improvements in healthcare systems and patient outcomes [1]. It provides critical evidence regarding the safety and efficacy of treatments and interventions, contributing to improved standards of care [2]. However, clinical research and healthcare activities can generate substantial carbon emissions, through a range of operational and research related activities.

Systematic reviews have estimated that global healthcare systems contribute approximately 4–5% of total greenhouse gas emissions [3]. A similar review, estimated emissions from healthcare systems accounted for between 1.5% − 9.8%, with a mean ratio of 4.9% [4]. Clinical research studies, like clinical trials, can generate significant emissions, for example the CASPS trial and PRIMETIME trials, produced and estimated 72 tonnes of $CO_2e$ (carbon dioxide equivalent) and 89 tonnes of $CO_2e$, respectively [5]. Emissions associated with clinical research arise from activities such as sample collection and storage, data management, participant and staff travel, and the use and disposal of laboratory materials [6].

The increasing contribution of greenhouse gas emissions to climate change presents a serious global challenge, with well documented impacts on environmental stability and extreme weather events [7]. Identifying and mitigating contributors to emissions across sectors including healthcare and research is therefore an urgent priority, including strategies to reduce them.

As healthcare systems globally commit to net-zero targets [8,9], understanding public awareness of sustainable research practices is increasingly important, as a key driver to implementing positive change in reducing healthcare - and research - related emissions. This increased understanding could influence education initiatives informing patients and the public about research design changes to improve sustainable research practices. Similarly, public agreement for more sustainable research practices could influence funding bodies and policymakers to implement standard reporting and/or cost adjustments to enable more environmentally friendly research. Finally, the adoption of emission reduction strategies and promotion of studies as 'lower carbon' could positively influence recruitment and participation. Key stakeholders, including patients and the public, are fundamental to the development of these practices. As such, this study will include public and patient involvement (PPI) partners, to support the co-development of a survey tool exploring public understanding of sustainable clinical research.

### Aim and objectives

The aim of this study is to co-design and co-develop a survey to investigate public understanding of sustainable clinical research. The objectives are to:

- Recruit PPI partners and co-develop survey themes, questions and format

- Distribute the co-developed online survey to members of the public

- Evaluate and report the PPI co-development process

• Analyse and report the survey responses

## Methods

This study comprises two phases outlined below and summarised in Fig 1.

### Phase I – Co-development of online survey

The survey will be co-developed through a structured process with PPI partners and is detailed below.

### Co-development process

The co-development process will involve two workshops with PPI partners. Proposed survey themes/topics will be provided in advance to support discussion (see Table 1).

During Workshop One, the survey topic and proposed themes, and initial generation of open – and closed – ended survey questions will be developed. Feedback from Workshop One in the form of transcripts from audio-recordings and written notes will be analysed and inform the drafting of survey questions and formatting for Workshop Two.

Workshop Two will focus on reviewing and finalising survey questions, content and gathering evaluative feedback on the co-development process. The researcher will then analyse transcripts and written notes from Worksop Two to draft the final survey and share with PPI partners for final feedback and approval. The final survey is anticipated to include five or six themes, with five questions per theme (25–30 questions). Approximate completion time is 30 minutes. However, based on question design, this may vary. PPI partners will then contribute to user testing before survey distribution.

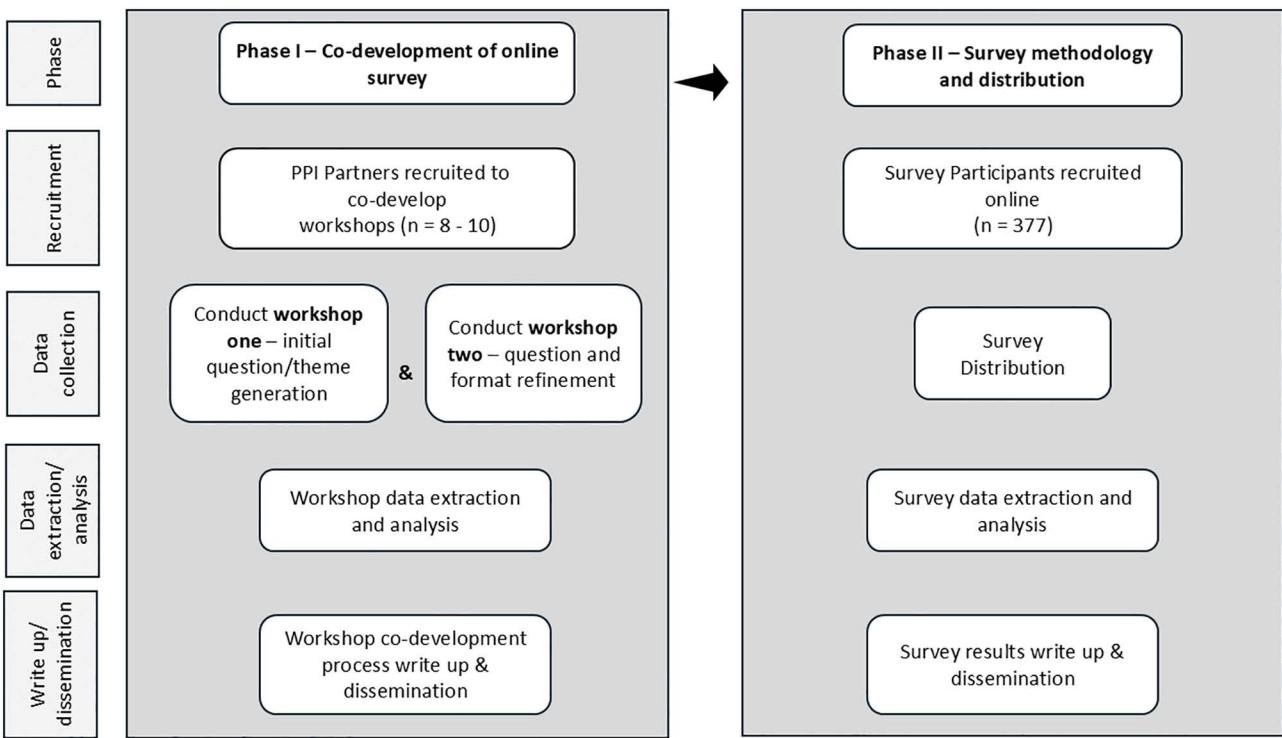

**Fig 1. Phase I and Phase II development processes.**

**Table 1. Proposed themes for discussion during workshop exercises.**

|   | Theme | Topics |
|---|-------|--------|
| 1 | Awareness and Understanding of Clinical Research | Knowledge of what clinical research is and how it works<br>Perception of the importance and benefits of clinical research |
| 2 | Willingness to Participate in Clinical Research | Factors that influence accessibility/decision-making (e.g., research location, use of digital tools, travel) |
| 3 | Instrument to assess environmental concern | Adapted environmental concern scale (to be agreed between researcher and PPI partners) |
| 4 | Environmental Concerns in Research | Views on reducing waste, energy use, travel emissions, and resource consumption in research<br>Level of concern about environmental impact from healthcare and research |
| 5 | Understanding of the environmental impact of clinical research | Awareness of environmental and social impact of clinical research<br>Perceptions of what makes research "environmentally friendly" |
| 6 | Equity and Access | Views on whether practices (e.g., remote participation) help or hinder inclusivity |

As well as the feedback from the workshop discussions being used to inform the survey design, the evaluation feedback will be analysed to report on the co-development process.

## Consensus process

A structured consensus approach will be used, drawing on a simplified 'card sorting' method. This method was traditionally deployed as a method to structure the navigation of web pages but has been adapted to priority-setting in healthcare research [10]. PPI partners will propose questions, which will be discussed, grouped into themes and prioritised collaboratively during the workshops. Multiple rounds of 'card sorting' may be used throughout the development process to finalise questions and design.

## Recruitment of PPI partners

PPI partners will be recruited through the University College Dublin Clinical Research Centre (UCD CRC) social media channels, the UCD PPI Ignite Network and if required, the National PPI Ignite Office. Experience in clinical research or sustainability will be desirable but not a requirement.

All PPI partners will be provided with an information leaflet and consent form for participation in the co-development research exercise (see S1 File PPI partner information leaflet and consent form).

## Sample size calculation

We will aim to recruit 8–10 PPI partners to support effective discussion and representation during the co-development process.

## Data extraction

Workshop transcripts will be imported into NVivo for analysis [11]. Two researchers will independently code a subset of the data to establish consistency and reduce analytical bias. Data will be analysed thematically in NVivo [11], using Braun & Clarke's (2006) thematic analysis framework [12]. This framework incorporates a six-step procedure for analysis of qualitative data:

- step one involves data familiarisation,

- step two involves generation of initial codes,

- step three involves generation of initial themes,

- step four involves reviewing themes,

- step five involves defining and finalising themes,

- step six involves generation of a report.

## Data analysis

Analysis of the co-development data may be subject to researcher bias. As such, measures will be taken to reduce bias and ensure inter-rater reliability. This includes the first researcher (DK) and a second researcher independently analysing a section of data as a pilot exercise, to identify discrepancies or issues in the analysis and coding process. A feedback loop and continuous monitoring will then be employed to address iteratively any deviations in coding [13]. The first researcher will code the remaining data. The researchers will then engage in peer debriefing, where they will meet to discuss the remaining data and associated codes to identify any substantial discrepancies [13,14]. This method will be applied across both workshops. As such, Workshop One data will inform the development of survey questions for discussion at Workshop Two. Workshop Two data will inform finalisation of questions and formatting, resulting in a final survey reviewed by PPI partners.

## User testing

PPI partners and peer colleagues will contribute to user testing of the survey prior to distribution. In particular, survey accessibility, access and navigation will be key considerations to ensure there is a high degree of usability.

## Phase II – Survey methodology and distribution

The online survey will be used to:

- Assess public knowledge and understanding of clinical research including awareness of environmentally sustainable research practices

- Explore public attitudes and beliefs regarding sustainability in clinical research

- Identify information needs related to sustainable research practices

The survey will be anonymous and hosted on Qualitrics as a secure, UCD approved, survey platform. Participants will be informed that survey responses will be anonymised and that no personally identifiable information (PII), including IP addresses and contact information will be collected. All data will be stored on UCD servers in accordance with institutional data governance policies. This will be enabled and communicated to participants.

## Survey participant population

Survey participants will comprise members of the public with links to clinical research networks, clinical research facilities, patient/charity or advocacy organisations associated to clinical research.

## Recruitment

Research participants will be recruited through multiple channels, including:

• clinical research facilities (CRFs)

• clinical trial networks (CTNs)

• clinical research centres (CRCs)

• public and patient involvement (PPI) networks

• national organisations, such as the Health Research Board – Trials Methodology Research Network (HRB-TMRN), Cancer Trials Ireland, and associate groups.

Colleagues within the above-mentioned networks will disseminate the survey link to potential participants. Direct contact between the researcher and survey with participants will not occur. A snowball sampling approach will be employed, whereby participants will be encouraged to share the survey link within their network [15].

In addition, the survey will be promoted via social media platforms and at public engagement events. A QR code and brief study description will be provided to enable access to the survey. To prevent duplicate responses or repeated participation, a feature in the survey software will be enabled that prevents multiple submissions.

## Informed consent

Prior to accessing the survey, participants will be provided with a participant information leaflet and consent form. Consent will be obtained by requiring participants to indicate agreement with a series of consent statements before proceeding to the survey questions (see S2 File survey participant information leaflet and consent form).

## Sample size

The survey sample has been estimated using the online Raosoft (www.raosoft.com/samplesize.html) calculator. Assuming a 5% margin of error, confidence level of 95% and response distribution of 50% were chosen, a minimum sample size of 377 respondents was estimated for large or unknown populations. Given the descriptive aims of this study and the use of non-probability sampling, the sample size is intended to provide a robust overview of public perceptions rather than population level estimates.

## Inclusion and exclusion criteria

Adults over the age of 18 with access to the internet will be eligible to participate. Non-eligible people include those under the age of 18, those that do not have access to the internet, or those who are non-English speakers, as the survey will be provided only in the English language.

## Strategies to minimise bias

Survey based research is subject to several potential sources of bias including response bias, sampling bias, non-response bias and order effects.

*Response bias* will be mitigated through the co-development of the survey content with PPI partners ensuring the language is simple and clear, accessible and avoids leading questions. The survey distribution approach will ensure a wide pool of potential participants are reached and respondents are anonymous which will further reduce risk of response bias.

*Sampling bias* will be addressed by distributing the survey through various clinical research and public facing networks and by using a snowball sampling approach to extend reach beyond initial distribution groups, increasing the random sampling. This approach may result in reducing representativeness in the survey population as it may exclude those that are not social connections, however the inclusion of networks with diverse connection pools, as well as adopting various distribution methods (email, social media, public engagement events) can address this.

*Non-response bias* will be minimised by designing the survey to be concise, accessible and compatible with screen reading software. As this is a web survey, non-response bias is unavoidable for those who experience difficulty with technology and online accessibility, however measures will be taken to ensure access for participants is as easy as possible.

*Order effects* will be reduced by randomising survey questions where appropriate, except where question order is required for logical coherence.

### Data extraction

Survey data will be exported from Qualtrics for analysis. Quantitative data will be analysed using descriptive statistics within Qualtrics. Qualitative free text responses will be analysed using inductive content analysis [16]. Automated text analysis tools (TextIQ) will be used to support data organisation in Qualtrics to code the data. This approach is suitable for analysing text-based data where you are not making pre-determined assumptions about the dataset, as well as data that is unknown or not previously researched. As this survey will generate new data regarding an unexplored topic, it is the preferred approach to take.

This approach differs from a deductive content analysis approach, where the researcher utilises predetermined codes and/or a previous conceptual framework to analyse data. Inductive content analysis takes a multi-step approach, including familiarising with the data, conducting multiple rounds of coding data iteratively, followed by synthesising and interpreting the data [17]. To reduce bias in this process, a peer analysis approach will be taken, where a second researcher will similarly code and interpret the data to ensure validity. If there are discrepancies between the codes generated, both researchers will discuss and resolve as appropriate. If required, a third researcher will be utilised to resolve discrepancies [13,14].

### Survey reporting

The checklist for reporting of survey studies (CROSS) will be used to guide transparent and standardised reporting of the survey findings [18]. The CROSS checklist (S3 File) comprises 40 items across 19 sections and was a checklist developed through consensus of experts, via a Delphi method [19].

### Study status

This study is at the recruitment stage for PPI partners.

### Study timeline

**PPI partner recruitment and workshop write up (estimated ~ 6 months).**

• Recruitment of PPI partners began 20/01/26

• Estimated recruitment will be completed by 31/03/26

• Estimated data collection will be completed by 15/05/26

• Estimated data analysis will be completed by 30/06/26.

Recruitment of PPI partners will take place for approximately one month. Two workshops will be held soon after over a one-month period. Following this, data analysis and write-up of data collected during the workshops will take place over

two to three months. A manuscript of the development workshop exercise will be written up over a two-month period and submitted for publication in a peer-reviewed open-access journal.

**Study participant recruitment and results write up (estimated ~5 months).**

• Estimated distribution of the survey will begin 31/05/26

• Estimated data collection will be completed by 30/09/26

• Estimated data analysis will be completed by 31/10/26.

The completed study survey will be distributed for 3 + months, or until the recruitment target (n = 377) is reached. Results from the survey will be analysed and written up for publication in a peer-reviewed open-access journal over a two-month period.

## Data management plan

Phase I data collected will include name and contact information for PPI partners, as well as data collected during the workshops. Data will be securely stored on UCD servers only accessible by the research team. Contact information will be stored for the duration of the study period and destroyed thereafter. Any potentially identifiable data collected during workshops will be anonymised.

Phase II survey data collected will be non-personal/non-identifiable data. If participants reach out to the research team via email for support with the survey, personal data will be securely stored on UCD servers, only accessible by the research team, and destroyed as soon as possible thereafter.

Anonymised study data will be archived for 10 years in accordance with university guidelines.

## Ethical considerations

All participants will be provided with clear, lay language information regarding the study prior to participation. Participants may withdraw by closing the survey, however due to the anonymous nature of data collection, once responses are submitted, they cannot be withdrawn. No personal data will be collected. Any personal information voluntarily provided via email correspondence with the research team will be deleted following resolution of the query.

Confidentiality and anonymity will be maintained throughout the study. Survey responses will be anonymised at the point of collection and all data will be stored securely on UCD servers.

## Ethics status

Ethics approval has been granted for this study by the University College Dublin (UCD) Human Research Ethics Committee (HREC) (reference LS-25–52-Keegan-Doran).

## Dissemination

Findings from this study will be disseminated through publication in an open access peer reviewed journal and presentations at relevant academic conferences and meetings. Results will also inform a subsequent larger study that is examining the sustainable nature of clinical research practices.

## Discussion

While research has been conducted in recent years to understand the environmental impact of clinical research activities, there is limited data and evidence of public knowledge and perceptions of sustainable research practices that could address this impact. Online survey research offers an increasingly common method for data collection of public perspectives, due to advances in survey software, data security, and online accessibility [20]. They facilitate efficient survey

design, data collection and analysis and may achieve favourable response rates compared with traditional survey methods particularly with larger samples sizes [21].

Development of this online survey will be strengthened by the inclusion of PPI, an established research approach, that will ensure rigour in survey instrument co-development, through co-design and user testing. Evidence suggests that PPI can enhance research relevance, quality and impact by incorporating lived experience and public perspectives [22,23]. Additionally, it will ensure that survey content, language and formats are accessible, meaningful and reflective of public priorities, rather than only academic or institutional perspectives.

While the online survey approach can introduce various biases and associated limitations, the study design will mitigate for these, specifically addressing sampling, response, and order effects concerns, through co-design, diverse distribution methods, and implementation of software features to improve readability and functioning.

The findings from this study will inform of public understanding of sustainable clinical research and inform the method and output from a PPI co-development process. Limited public awareness may indicate a need for improved communication and education, whereas strong public concern could support prioritisation of investment in more sustainable research methodologies and infrastructure. The resulting findings may also inform institutional policies, funding decisions, ethical guidelines, and sustainability strategies within research organisations. Additional policy implications centred on research funders could influence adoption of routine environmental impact assessments and/or carbon budgeting for studies. Finally, the findings will further inform future research activities for a larger study looking at the development of sustainable research practices.

## Supporting information

**S1 File. PPI partner information leaflet and consent form.**
(PDF)

**S2 File. Survey participant information leaflet and consent form.**
(PDF)

**S3 File. CROSS checklist.**
(PDF)

## Author contributions

**Conceptualization:** Dylan Keegan, Peter Doran.

**Funding acquisition:** Peter Doran.

**Methodology:** Dylan Keegan, Linda O'Neill, Peter Doran.

**Project administration:** Dylan Keegan.

**Supervision:** Linda O'Neill, Peter Doran.

**Writing – original draft:** Dylan Keegan.

**Writing – review & editing:** Dylan Keegan, Linda O'Neill, Peter Doran.

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
