## [Decision Letter · Decision Letter 0]

23 Mar 2026

PONE-D-26-02758Co-developing a Survey on Public Understanding of Sustainable Clinical Research: A Study ProtocolPLOS One

Dear Dr. Keegan,

Thank you for submitting your manuscript to PLOS ONE. After careful consideration, we feel that it has merit but does not fully meet PLOS ONE’s publication criteria as it currently stands. Therefore, we invite you to submit a revised version of the manuscript that addresses the points raised during the review process.

**ACADEMIC EDITOR: I congratulate you on your perseverance in the review process. However, to make your manuscript more robust we need to address few more comments from the reviewers'.**

We look forward to receiving your revised manuscript.

Kind regards,

Tanay Chaubal

Academic Editor

PLOS One

2. Please include a separate caption for each figure in your manuscript.

Reviewers' comments:

Reviewer's Responses to Questions

**Comments to the Author**

1. Does the manuscript provide a valid rationale for the proposed study, with clearly identified and justified research questions?

Reviewer #1: Yes

Reviewer #2: Yes

2. Is the protocol technically sound and planned in a manner that will lead to a meaningful outcome and allow testing the stated hypotheses?

Reviewer #1: Yes

Reviewer #2: Yes

3. Is the methodology feasible and described in sufficient detail to allow the work to be replicable?

Reviewer #1: Yes

Reviewer #2: Yes

4. Have the authors described where all data underlying the findings will be made available when the study is complete?

Reviewer #1: Yes

Reviewer #2: Yes

5. Is the manuscript presented in an intelligible fashion and written in standard English?

Reviewer #1: Yes

Reviewer #2: Yes

6. Review Comments to the Author

You may also provide optional suggestions and comments to authors that they might find helpful in planning their study.

Reviewer #1: This manuscript presents a well-designed and timely study protocol addressing the increasingly important issue of sustainability in clinical research. The integration of Public and Patient Involvement (PPI) in the co-development of the survey instrument is a notable methodological strength that enhances relevance, inclusivity, and rigor. The study design is clearly structured, ethically sound, and appropriately aligned with established qualitative and survey reporting frameworks. The manuscript would benefit from minor clarifications regarding the long-term data-sharing plan and a slightly expanded discussion on sampling limitations and policy implications. These refinements will improve transparency and strengthen the overall impact of the work. Subject to these minor revisions, the manuscript is suitable for publication.

Reviewer #2: This manuscript presents a study protocol describing the co-development and deployment of a survey designed to evaluate public understanding of environmentally sustainable clinical research. The study integrates Public and Patient Involvement (PPI) through a structured co-development process and aims to generate insights into public awareness, perceptions, and attitudes toward sustainability in clinical research practices.

Overall, the protocol is clearly structured and well aligned with contemporary approaches to participatory research and sustainability in healthcare. The manuscript would benefit from minor clarifications and improvements in presentation prior to publication.

Clarity of Study Rationale:

The introduction clearly outlines the environmental impact of healthcare and clinical research. However, the rationale for focusing specifically on public understanding could be strengthened by briefly explaining how such understanding may influence policy, research practices, or participation in sustainable research initiatives.

Description of Survey Instrument Development:

The manuscript describes the co-development process using two workshops with PPI partners. While the process is well explained, it may be helpful to briefly describe how the final survey will be structured (for example, approximate number of questions or estimated completion time) to give readers a clearer picture of the resulting instrument.

Recruitment Strategy:

The protocol outlines several recruitment channels including clinical research networks, patient organisations, and social media distribution. A brief clarification of how duplicate responses or repeated participation will be prevented would improve methodological transparency.

Bias Considerations:

The manuscript appropriately acknowledges potential survey biases such as response bias, sampling bias, and non-response bias. Expanding briefly on how snowball sampling may influence the representativeness of the sample would strengthen this section.

7. PLOS authors have the option to publish the peer review history of their article (what does this mean?). If published, this will include your full peer review and any attached files.

Reviewer #1: No

Reviewer #2: **Yes:** Akshay Ashok Katara

---

## [Author Response · Author response to Decision Letter 1]

1 Apr 2026

Editor

We thank the editor for the additional information regarding journal requirements. The manuscript and files have been formatted to meet the journal’s style requirements. This includes edits to the title page, citation formatting throughout, headings throughout, the reference list, and supporting information section.

Reviewer #1: This manuscript presents a well-designed and timely study protocol addressing the increasingly important issue of sustainability in clinical research. The integration of Public and Patient Involvement (PPI) in the co-development of the survey instrument is a notable methodological strength that enhances relevance, inclusivity, and rigor. The study design is clearly structured, ethically sound, and appropriately aligned with established qualitative and survey reporting frameworks. The manuscript would benefit from minor clarifications regarding the long-term data-sharing plan and a slightly expanded discussion on sampling limitations and policy implications. These refinements will improve transparency and strengthen the overall impact of the work. Subject to these minor revisions, the manuscript is suitable for publication.

We thank reviewer 1 for their valuable feedback. Clarification has been added regarding the long-term data-sharing plan under the heading ‘Data management plan’. The discussion has been expanded to include points on limitations related specifically to the snowball sampling approach and how the authors will address this. Additionally, further detail regarding influence on policy implications has been added.

Reviewer #2: This manuscript presents a study protocol describing the co-development and deployment of a survey designed to evaluate public understanding of environmentally sustainable clinical research. The study integrates Public and Patient Involvement (PPI) through a structured co-development process and aims to generate insights into public awareness, perceptions, and attitudes toward sustainability in clinical research practices.

Overall, the protocol is clearly structured and well aligned with contemporary approaches to participatory research and sustainability in healthcare. The manuscript would benefit from minor clarifications and improvements in presentation prior to publication.

We thank reviewer 2 for their valuable feedback that has been addressed below.

Clarity of Study Rationale:

The introduction clearly outlines the environmental impact of healthcare and clinical research. However, the rationale for focusing specifically on public understanding could be strengthened by briefly explaining how such understanding may influence policy, research practices, or participation in sustainable research initiatives.

The introduction has been revised to include additional detail regarding the influence of public understanding on education initiatives, standardised environmental reporting and cost adjustments, as well as the potential influence on future participant recruitment and participation.

Description of Survey Instrument Development:

The manuscript describes the co-development process using two workshops with PPI partners. While the process is well explained, it may be helpful to briefly describe how the final survey will be structured (for example, approximate number of questions or estimated completion time) to give readers a clearer picture of the resulting instrument.

As the survey design is contingent on the co-development process it is unclear of the exact number of survey questions to be included. However, clarification has been added to note there are six potential themes for inclusion and we would expect to limit each theme to five questions. The estimated survey completion time should be 30 minutes. Once the survey has been finalised and pilot tested, the estimated completion will be updated and added to the survey brief.

Recruitment Strategy:

The protocol outlines several recruitment channels including clinical research networks, patient organisations, and social media distribution. A brief clarification of how duplicate responses or repeated participation will be prevented would improve methodological transparency.

This is an important consideration given the nature of the recruitment approach.

A clarification has been added to note that a feature of the survey software can be enabled to prevent multiple submissions.

Bias Considerations:

The manuscript appropriately acknowledges potential survey biases such as response bias, sampling bias, and non-response bias. Expanding briefly on how snowball sampling may influence the representativeness of the sample would strengthen this section.

Clarification has been added regarding the challenge of achieving representativeness from a snowball sampling approach. The distribution method can be potentially limited, however the authors believe the chosen networks and communities on the distribution list will address this sufficiently.

---

## [Decision Letter · Decision Letter 1]

27 Apr 2026

Co-developing a survey on public understanding of sustainable clinical research: a study protocol

PONE-D-26-02758R1

Dear Dr. Dylan Keegan,

We’re pleased to inform you that your manuscript has been judged scientifically suitable for publication and will be formally accepted for publication once it meets all outstanding technical requirements.

Kind regards,

Tanay Chaubal

Academic Editor

PLOS One

Additional Editor Comments (optional):

Reviewers' comments:

Reviewer's Responses to Questions

**Comments to the Author**

1. Does the manuscript provide a valid rationale for the proposed study, with clearly identified and justified research questions?

Reviewer #2: Yes

2. Is the protocol technically sound and planned in a manner that will lead to a meaningful outcome and allow testing the stated hypotheses?

Reviewer #2: Yes

3. Is the methodology feasible and described in sufficient detail to allow the work to be replicable?

Reviewer #2: Yes

4. Have the authors described where all data underlying the findings will be made available when the study is complete?

Reviewer #2: Yes

5. Is the manuscript presented in an intelligible fashion and written in standard English?

Reviewer #2: Yes

6. Review Comments to the Author

You may also provide optional suggestions and comments to authors that they might find helpful in planning their study.

Reviewer #2: Thank you for submitting the revised version of this manuscript. The authors have responded thoroughly and professionally to the prior reviewer comments. The revisions have significantly improved the manuscript’s clarity, transparency, and methodological completeness. The study protocol now meets the expectations for publication as a well-designed and timely contribution in the area of sustainable clinical research and public engagement.

7. PLOS authors have the option to publish the peer review history of their article (what does this mean?). If published, this will include your full peer review and any attached files.

Reviewer #2: **Yes:** Akshay Ashok Katara

---

## [Editor Report · Acceptance letter]

PONE-D-26-02758R1

PLOS One

Dear Dr. Keegan,

I'm pleased to inform you that your manuscript has been deemed suitable for publication in PLOS One. Congratulations! Your manuscript is now being handed over to our production team.

Kind regards,

on behalf of

Dr. Tanay Chaubal

Academic Editor

PLOS One